# FDG-PET to T1 Weighted MRI Translation with 3D Elicit Generative Adversarial Network (E-GAN)

**DOI:** 10.3390/s22124640

**Published:** 2022-06-20

**Authors:** Farideh Bazangani, Frédéric J. P. Richard, Badih Ghattas, Eric Guedj

**Affiliations:** 1Department of Mathematics and Computer Science, CNRS, Aix Marseilles University, UMR, 7249 Marseille, France; farideh.bazangani@fresnel.fr (F.B.); badih.ghattas@univ-amu.fr (B.G.); 2Molecular Neuroimaging, Marseille Public University Hospital System, 13005 Marseille, France; eric.guedj@fresnel.fr

**Keywords:** deep learning, generative adversarial network, medical image synthesis

## Abstract

Objective: With the strengths of deep learning, computer-aided diagnosis (CAD) is a hot topic for researchers in medical image analysis. One of the main requirements for training a deep learning model is providing enough data for the network. However, in medical images, due to the difficulties of data collection and data privacy, finding an appropriate dataset (balanced, enough samples, etc.) is quite a challenge. Although image synthesis could be beneficial to overcome this issue, synthesizing 3D images is a hard task. The main objective of this paper is to generate 3D T1 weighted MRI corresponding to FDG-PET. In this study, we propose a separable convolution-based Elicit generative adversarial network (E-GAN). The proposed architecture can reconstruct 3D T1 weighted MRI from 2D high-level features and geometrical information retrieved from a Sobel filter. Experimental results on the ADNI datasets for healthy subjects show that the proposed model improves the quality of images compared with the state of the art. In addition, the evaluation of E-GAN and the state of art methods gives a better result on the structural information (13.73% improvement for PSNR and 22.95% for SSIM compared to Pix2Pix GAN) and textural information (6.9% improvements for homogeneity error in Haralick features compared to Pix2Pix GAN).

## 1. Introduction

In some medical image applications like segmentation, classification, and detection, the lack of database and unbalanced data (biased on age, normal subject, or abnormal subject) can affect the accuracy of deep learning models. Some studies tried to overcome this issue with traditional data augmentation methods. The most common data augmentation method includes simple transformations of the dataset such as rotation, crop, scale, etc. Although these methods improve the training process in a deep network, the informative information which is provided for the network is not significant [1]. High-quality synthetic data augmentation with generative models can enrich the dataset in terms of variability.

There are two different tasks in medical image synthesis: image generation and image-to-image translation or cross-modality generation. In a cross-modality generation, we transfer a source domain into a target domain. In image generation, the model generates a specific target domain from noise. In this study, we focus on image-to-image translation.

Most neuroimages are 3D images such as computed tomography (CT), magnetic resonance imaging (MRI), and positron emission tomography (PET). Therefore for processing these images, 3D operations are necessary to capture spatial information in all the dimensions. Processing 3D images is a hard task for both image generation and image-to-image translation. The complexity and computational cost of 3D synthesis is high compared to 2D. Patching images can decrease the computation but missing the global structural information after patching is a drawback for this strategy.

Another challenge in medical image synthesis is to evaluate the quality of generated images. There are two typical evaluation methods: task-dependent evaluation and intrinsic evaluation. In the task-dependent evaluation, the evaluation of the images is done based on the accuracy enhancement in another task (segmentation, classification and…) rather than the quality of each individual image [2]. The intrinsic evaluation compares the simulated data with the ground truth in a supervised simulation. The task-dependent evaluation assessment cannot provide information on the texture properties or structure of the simulated images. Normally the intrinsic evaluation is obtained by matrices that focus on the voxel values and the quantity of noise in the generated images. In medical images, the texture, the structure, and distribution of voxel values are essential information for image processing. Therefore, evaluating the simulation method in terms of texture and structural analysis is relevant.

In this study, we present a cross-modality generation method to generate T1-weighted MRI from Fluorine-18-Flourodeoxygloccose Positron Emission Tomography (FDG-PET). This dual information (PET + MRI) is not always available, especially in the same temporal window (If the two examinations are not done at the same time, the two parameters change over time with a loss of information). Having the dual modalities (PET + MRI) for the inclusion of healthy subjects in scientific studies is beneficial. Besides these aspects, the map of grey matter density will allow correcting the partial volume effect on PET images and thus improve the quality of PET images. Sometimes in diagnosing, some brain diseases having dual modalities is useful to improve the accuracy of the diagnosis [3]. The structural and functional information of brain tissues in MRI can be mapped from PET images, but it is hard to map the structure information such as the skull [3]. Despite this, the objective is not to entirely replace MRI with PET since MRI is multimodal imaging not limited to the measure of grey matter density. In this study, we focused on mapping the information of brain tissues, especially the grey-matter density for healthy subjects to simplify the mapping.

We propose a cross-modality generation method, called Elicit generative adversarial network (E-GAN). To reduce the complexity and number of parameters in a 3D network, we proposed a separable convolution network to elicit the relevant features of FDG-PET. Then 2D high-level features were concatenated with the 3D positional information which is retrieved from a Sobel filter to reconstruct the 3D weighted MRI image by a decoder. The main contributions of this paper are as follows: (I) We split learning 3D features with separable convolution in a generative model instead of full convolutions. (II) We propose a fusion strategy similar to the self-attention mechanism in order to put in order the extracted features. (III) E-GAN includes a Sobel filter for transmission of the geometrical information instead of using 3D convolutions. (VI) We improved the stability of learning with a weighted version of a hybrid loss function.

The rest of the paper is organized as follows. In Section 2 we describe the related work. In Section 3, we introduce the theory related to E-GAN. Then, we describe the framework of our proposed model in Section 4. Next, we present the experimental settings, experimental results, and discussion in Section 5. Finally, we conclude the paper in Section 6.

## 2. Related Work

Early works for translating medical images focused on statistical approaches based on voxel-based estimation and mappings [4]. In one of the earliest machine learning approaches a random forest has been proposed for synthesizing standard-dose brain [18F] FDG-PET images from the low-dose brain [18F] FDG-PET and T1 weighted MRI [5]. Since there is much uncertainty and variability in some neuroimaging modalities like FDG-PET, most of the time, these approaches are not efficient.

In the meantime, by developing the strengths of deep learning and generative models many studies focused on deep learning methods for image-to-image translation in 2D. One of the primary works on the application of deep learning was a 2D variational auto-encoder [6]. Taking an MRI modality (e.g., T1, T2, VFlair, or DWI) as an input, a deep Encoder-Decoder Image Synthesizer (DEDIS) simulates as an output an MRI of another modality with the same size as the input.

After Goodfellow et al. proposed the generative adversarial network (GAN) [7], many researchers started to synthesize medical images with GAN. This model is composed of two networks which are a generator and a discriminator interacting through an adversarial process. In [8], Chartsias et al. suggested one of the first GAN-based model for an unpaired medical image-to-image translation. This approach uses an adversarial training to learn to transform CT images to MRI. Both modalities are aligned to prevent the network from learning the structural differences instead of the intensity statistics.

In another study, Salman et al. designed a method to synthesize a multicontrast MRI from a 2D conditional GAN [9]. To improve the accuracy, they suggested taking into account additional information from correlated structures across neighboring cross-sections. This method improved the quality of the generated images, particularly when the source images contain a significant amount of noise.

A 2D GAN was proposed in [10] called multistream GAN (MustGAN). This architecture holds information from various source contrasts by adaptively joining one-to-one to many-to-one streams. In this model, there are K one-to-one stream networks and a many-to-one stream network. Each one-to-one network has a unique feature map that is shared with the many-to-one stream network to synthesize the target contrast image.

In most of the studies, medical image synthesis or image-to-image translation has been done by synthesizing slices and therefore neglecting the spatial information in 3D. The mildest strategy to generate 3D images is considering each axial slice independently from sagittal and coronal. With such a model, 2D slices are synthesized and then stacked to obtain a 3D image.

In [11], Zhou et al. proposed a hybrid fusion network to synthesize MRI from different modalities. They performed a modality-specific network to capture high-level features from different modalities. Then, they computed adaptive weights to include these features into the synthesis network. The suggested architecture uses 2D axial-plane slices of the volumes and generates 2D slices of MRI images.

In [12], an MRI synthesis was achieved with a dual discriminator adversarial learning which includes a difficulty-aware attention mechanism. This mechanism aims to improve the generation of hard-to-synthesis voxels in the hippocampus regions. The generated images are then used to achieve a better accuracy in a hippocampal sub-fields segmentation. The 3D images are obtained from stacking 2D images in the third axis and fed to a 3D convolution network for segmentation. The stacking approach reduces the synthesis quality of the volume, misses the information in one axis and causes discontinuities in the third axis.

To reduce the problem of discontinuities, a 3D conditional GAN is proposed in [13,14]. In [14] the authors focused on improving the accuracy of a tumour segmentation by synthesizing FLAIR images from T1 weighted MRI. They proposed a 3D conditional Generative Adversarial Network (cGAN) with a U-net architecture and a local adaptive fusion method. In the local adaptive fusion method, they linearly combined the estimated images with the real images. The combination weights are estimated from the output of the 3D cGAN. The local adaptive fusion method can be implemented as a neural network to estimate the weights during the training. Wang et al. proposed a 3D auto-context-based locality adaptive multimodality generative adversarial networks model (LA-GAN) [13]. The locality adaptive network in this study is an end-to-end trained deep network and the fusion weights are automatically and simultaneously learned with the generative model. The objective of [13] is to synthesize high-quality PET images from the low-dose images in order to reduce the radiation exposure. The locality adaptive fusion network is fed with different modalities and generates the fused images. Then, the 3D U-net generates FDG-PET from the fused images. The training dataset contains 20 subjects. After patching images the size of the dataset has been increased to 2500 samples. Even though data augmentation can help the network to prevent overfitting, training a deep architecture like U-net with a few samples does not allow the model to be generalized.

In [13,14], the authors rely upon minimizing pixel/voxelwise intensity variations and ignore the textural details of the image content structure. Contextual information such as edge similarity could help the network improve the performance of the synthesizing. In [15], the authors proposed an edge-aware generative adversarial network (Ea-GAN) with a U-Net architecture for the cross-modality MR image synthesis in 3D. During training, a combination of a voxelwise intensity similarity and an edge similarity was minimized. Furthermore the edge maps have been integrated to improve the performance of the synthesis. To fix the saturation issue in the network, they used two strategies. The first strategy is label smoothing in the discriminator to increase the difficulty of differentiation and reduce the vulnerability of adversarial learning. In the second strategy they gradually increased the weight related to the expectation of the edge similarity in order to balance the importance of the edge information. The second strategy has been applied for a better use of the edge information in MR images. Due to the the synthesis cost and the number of parameters in the U-net architecture they proposed to patch the images with an overlap.

Recently, Lin et al. [3] proposed a 3D reversible GAN for a bidirectional mapping of 3D MRI and PET. This mapping has been done through a reversible network in order to transfer the information from one modality to another. A reversible network [16] is an invertible architecture that is practical when we have missing data in both the target domain and the observed domain.

Adapted image-to-image translation conditional GAN (pix2pix [17]), was designed to produce synthetic abnormal brain tumors in multiparametric MRI from corresponding segmentation masks [18]. In order to reduce the complexity of the model the images were axially cropped. Therefore the model was fed with 108 central slices and then the central part re-sampled into (128 × 128 × 54) which caused the loss of some small tumours.

Most of the 3D studies used U-net architecture [19,20,21]. U-net is a powerful architecture in medical image analysis. However implementing U-net with the min–max strategy causes convergence issues because of the complexity of the network in 3D. Toga et al. [22] proposed a self-attention mechanism with a feature matching loss in a U-net in order to improve the stability of training in 3D GAN. Table 1 represents a comparison between the state of the art methods and the proposed method.

According to Table 1, most of the studies propose a complex network for 3D generation. Training a complex model needs more data which is a challenge in medical images. Lack of data and the need of a 3D network in order to obtain spatial information makes generating 3D neuroimage a challenging research topic with many open questions.

## 3. Generative Adversarial Network

A generative adversarial network (GAN) is composed of two networks: a generator and a discriminator that are combined using an adversarial process. Vanilla GAN is the original variant of GAN for the synthesis of artificial images proposed by Goodfellow and colleagues [7]. In Vanilla GAN the generator takes samples *z* from a noise distribution p(z) as the input and generates the output $x_{g}=G\left(z\right)$. The real samples $x_{r}$ from real data distribution p(x) and the generated sample $x_{g}$ from the generator are used as the input of the discriminator *D* to produce a single value which is the probability to be real or fake.

During training, the discriminator provides feedback for the generator through a positive or negative gradient related to the quality of the generated image by the generator. The discriminator *D*, which is a binary classifier, attempts to distinguish the real images from the fake ones generated by *G* while the generator *G* tries to fool *D* by generating realistic examples. The opposition between the two networks, *D* and *G*, can be formalized as a min–max game. The minimization and maximization parts used to train the generator and the discriminator that represented by the loss function as follows [7]:$$min_{G}max_{D}L\left(D,G\right)=E_{x∼p_{data(x)}}\left[log\left(D\left(x\right)\right)\right]+E_{y∼p_{z}\left(z\right)}\left[1-log\left(D\left(G\left(z\right)\right)\right)\right]$$

GAN is used to generate new samples having the same distribution as the dataset using a random noise as the input. For an image-to-image translation task, conditional GAN (cGAN) is more suitable. cGAN takes an additional input y (labels, text, image, …) sampled from an empirical distribution. The loss function for training cGAN is defined as:$$min_{G}max_{D}L\left(D,G\right)=E_{x∼p_{data(x)}}\left[log\left(D\left(x|y\right)\right)\right]+E_{y∼p_{z}\left(z\right)}\left[1-log\left(D\left(G\left(z|y\right)\right)\right)\right]$$

During training, if one of the networks faces difficulties, the other network may fail to converge. Sometimes the generator feeds the discriminator with a frequent pattern. Consequently, the discriminator is not able to provide an informative feedback for the generator. As a result, the generator and the discriminator play a never-ending mouse-and-cat game [23].

Another common training difficulty occurs at the start of the training when the discriminator becomes too strong with a high accuracy to discriminate the fake images from the real ones. The discriminator is not able to provide feedback for the generator when 1−log(D(G(z)))≃0, so the learning process stops for the generator [24].

According to [25], the min–max strategy minimizes the Kullback-Leibler (KL) divergence $KL(P_{r}∥P_{g})$. To be able to define this distance the distribution of $P_{g}$ should exist. When such a distance does not exist, the algorithm might not converge.

Although the min–max strategy has many challenges in practice, if it converges, the quality of the generated image is better compared to the alternative approaches. In the next section, we introduce E-GAN with a min–max strategy for synthesizing 3D MRI in an image-to-image translation task.

## 4. Method

Our aim is to simulate the T1-weighted MRI from the corresponding FDG-PET image with Elicit generative adversarial network (E-GAN). This is obtained by training a generator *G* mapping the space *P* of PET images into the space *M* of MRI.

This mapping has been done by a generative adversarial network. The discriminator is a 3D fully convolutional network and The generator consists of two parts: an encoder and a decoder. The encoder has two parts: an Elicit network and a Sobel filter. The main goal of the encoder is to map the informative information of PET images with a shallow network. Then, the decoder generates the T1 weighted MRI from the retrieved information from PET images. How well the translation G(p) fits to *M* is evaluated by a loss function including two terms, Jensen Shannon distance (JSD) and $L_{2}$ loss. We trained the network on FDG-PET images and pairwise T1 weighted MRI with 256 subjects. In this section, we explain E-GAN in three parts: the network configuration, loss function, and training.

### 4.1. Network Configuration

The first part of the generator is a encoder network called Elicit network to extract the spatial information from 3D FDG-PET and map the features into 2D with a shallow network. The second part is a 3D decoder to generate the T1 weighted MRI form the extracted features (Figure 1). For the Elicit network, we were inspired by the Multi-view Separable Pyramid Network [26]. The main aim of the Elicit network is to obtain the spatial information from PET images with less complexity in the network. Separable convolution [27] divides up the spatial dimensions of an image with less multiplication. Therefore, the separable strategy can efficiently reduce the complexity in comparison with a 3D convolution.

As represented in Figure 2, we used two blocks of separable convolutions in sagittal, coronal, and axial views to map the spatial information in each axes of the PET image into 2D. Each block includes three separable convolutions. The first separable convolution is of size (90, 1, 1) for axial, (1, 90, 1) for sagittal, and (1, 1, 90) for coronal view. These blocks capture long-range dependencies in each axis. The second part of Elicit network has two separable convolutions of size (45, 1, 1), (1, 45, 1), and (1, 1, 45) for axial, sagittal and coronal views respectively. These blocks with a smaller kernel size capture local dependencies and information in the PET images. After the separable convolutions we have 2D spatial information extracted from 3D images. The second part of the Elicit network includes two 2D convolutions to extract the related features in each axis. Figure 2, represents the Elicit network in detail.

After projecting the spatial information we obtained the feature maps of each axis in 2D. The 2D features are represented as $x\in R^{\left(c\times c\right)}$ for sagittal view, $y\in R^{\left(c\times c\right)}$ for coronal view and $z\in R^{\left(c\times c\right)}$ for axial view. The parameter *c* represents the size of the feature map after the last layer in the Elicit network.

In order to mix these dependencies, we have implemented a fusing strategy similar to the self-attention mechanism [28] without the convolutions to combine long-range dependencies. Therefore for mixing the features, we transformed the axial feature maps and the transpose of the sagittal feature maps into *g*. The feature maps of the coronal axis are also transformed with the feature space *g* into *l*. In summary, the elementwise product of each two axes has been calculated. Then in order to limit the magnitude, the hyperbolic tangent of the features has been computed. At the end, we normalized the mixed feature maps. The third axis has been added with the same strategy to $g^{\left(i,j\right)}$. This transformation has been done by a Hadamard product as follows:$$g^{\left(i,j\right)}=\frac{\tanh x^{i,j}⊙{\bar{y}}^{i,j}}{\Sigma_{i=1}^{N}\tanh x^{i,j}⊙{\bar{y}}^{i,j}}$$
*N* represent the number of features on each axis. The output of mixing the two axes will merge with the third axis as follows:
$$l^{\left(i,j\right)}=\frac{\tanh g^{i,j}⊙{\bar{z}}^{i,j}}{\Sigma_{i=1}^{N}\tanh g^{i,j}⊙{\bar{z}}^{i,j}}$$

By encoding the high-level features with the Elicit network and the fusion mechanism, the model can only capture the non-geometrical information. In this basic form, it would misplace the geometrical information for decoding the MRI images. Hence, to transfer the geometrical information, we completed the Elicit network with a 3D Sobel filter [29].

Sobel filter detects the edges and transfer the information of the boundaries between different tissues of the brain for mapping the geometrical representation [30]. In addition, by adding this geometrical information, the generator starts with good knowledge from the PET image and generates more realistic samples. With this method, the generator can fool the discriminator in the first steps of training so the model faces the saturation problem less often.

Figure 3, represents the 3D Sobel operator, x, y and z axes correspond to each axis.

We can define the 3 × 3 × 3 filter as follows: $$G_{z}^{{}^{\prime }}\left(:,:,-1\right)=\left[\begin{array}{ccc} 1 & 2 & 1\\ 2 & 4 & 2\\ 1 & 2 & 1 \end{array}\right]$$
$$G_{z}^{{}^{\prime }}\left(:,:,0\right)=\left[\begin{array}{ccc} 0 & 0 & 0\\ 0 & 0 & 0\\ 0 & 0 & 0 \end{array}\right]$$
$$G_{z}^{{}^{\prime }}\left(:,:,1\right)=\left[\begin{array}{ccc} -1 & -2 & 1\\ -2 & -4 & -2\\ -1 & -2 & -1 \end{array}\right]$$

By performing Sobel filter the extracted edges are scale dependent. To fix this issue, we applied a 3D convolution after Sobel operator to have the same scale as the feature map and smooth the thickness of some edges. The last part of the generator decodes the T1-weighted MRI from high-level features and the geometrical information retrieved from the FDG-PET images. The decoder consists of four layers of 3D convolutions following a Batch normalization and a LeakyReLU as the activation function (Figure 1).

The input of the decoder is composed of two channels, one is the 2D feature maps of the PET images and the second channel is relevant geometrical information and the edges. The generator at the end generates the 3D T1-weighted MRI from 3D PET images with the proposed architecture.

### 4.2. The Loss Function

An adversarial loss minimizes the distance between the probability density function of the generated data and the real data but in practice it might be difficult to minimize. In order to reduce the problems of adversarial loss we implemented a loss function consisting of two terms: adversarial loss and $L_{2}$ loss. The $L_{2}$ loss, regardless of the structure in the image, deals with the large errors, but it is more tolerant to the slight errors [31]. Since we are dealing with the image-to-image translation task the adversarial loss is defined as follows:$$L_{\mathrm{GAN}}\left(G,D\right)=E_{\left(p,m\right)}\log\left(D\left(p,m\right)\right)+E_{p}\log\left(1-D\left(p,G\left(p\right)\right)\right)$$

As suggested in [32] we minimized the Jensen-Shannon divergence (JSD) to generate more realistic samples. The $JSD(p_{g}∥p_{r})$ is the symmetric version of KL divergence between $p_{g}$ which is the probability distribution of generated data and $p_{r}$ which is the probability distribution of the real data.
$$JSD\left(p_{g}∥p_{r}\right)=\frac{1}{2}KL\left(p_{g}∥\left(\frac{p_{r}+p_{g}}{2}\right)\right)+\frac{1}{2}KL\left(p_{r}∥\left(\frac{p_{r}+p_{g}}{2}\right)\right)$$

In practice, we improved the learning stability with a weighted version of the loss:
$$\arg\min_{G}\left(\arg\max_{D}\alpha_{1}L_{\mathrm{GAN}}\left(G,D\right)+\alpha_{2}L_{2}\left(G\right)\right)\;p\in P,m\in M$$
we considered $\alpha_{1}=1$ and $\alpha_{2}=0.5$.

### 4.3. Training

We trained the network with a mini-batch stochastic gradient descent and an Adam optimizer solver with a learning rate of 0.0001. The coefficients used for the computing running averages of the gradient and its square $B=(\beta_{1},\beta_{2})$ have been set to 0.5 and 0.999 respectively to have a stable training [33]. We initialized the weights with a normal distribution N(0.0,0.02). All the training was carried out on a single Nvidia GeForce RTX 3090 Ti with a batch size of 12 for 1000 epochs.

## 5. Experimental Results and Evaluation

To assess the performance of E-GAN for multimodality image translation, we conducted experiments on the Alzheimer’s Disease Neuroimaging Initiative (ADNI) dataset. The FDG-PET images and the pairwise T1 weighted MRI of 256 healthy subjects (129 female and 127 male) have been used for this study. In ADNI participants usually take some scans at different periods of time, we only used the subjects with less than one year delay between the scans. By adding this condition we could attenuate the differences caused by a long time delay between successive acquisitions.

Both the MRI and PET images were first registered in the Montreal Neurological Institute (MNI) space with spatial normalization to have the same resolution, with a voxel size of 2×2×2 mm${}^{3}$. Then an intensity normalization was performed on the PET images to divide each voxel intensity by the global average value. Eventually, to prepare the images to feed the network a voxel-normalization between [−1,1] and an interpolation with anti-aliasing to down-size the images into (90, 90, 90) have been implemented. All the procedures before the voxel-normalization have been done in SPM12 [34].

We evaluated the E-GAN with 3 models, a 3D DCGAN [35], a 3D WGAN [25], and a 3D Pix2Pix GAN [17]. The 3D DCGAN has been implemented following [35]. The architecture of the WGAN [25] has been designed to be the same as 3D DCGAN. The pix2pix GAN has been designed based on [17] with a 3D architecture. Training and optimization parameters have been experimentally set according to the stability of the minimization procedure and the quality of the generated images.

We evaluated the model in two different ways. First, we assessed the similarity between the simulated and the real images visually and by computing common evaluation metrics. We further proposed a structural analysis for the simulated images. This evaluation takes into account both the boundaries and the texture of the synthesized images.

### 5.1. Visual Image Similarity

Figure 4 represents a FDG-PET sample, a ground truth MRI image, and the generated image. The generated T1 weighted MRI, visually is quite the same as the ground truth. In order to compare the result with the state of the art methods Figure 5, represents a visual comparison between the proposed model and the three methods in the state of the art.

According to Figure 5, WGAN and DCGAN are defeated by Pix2Pix GAN and the proposed model in terms of visual assessment. The proposed method has higher quality and almost no artifacts and blurriness compared to Pix2Pix GAN.

The DCGAN almost could not generate the uniform tissues and the generated image only has the main structure of the T1 weighted MRI. While the synthetic result of the WGAN has a better performance on synthesizing the structure of T1 weighted MRI, the synthetic images often preserve noise. The Pix2Pix GAN architecture obtains more details inside the tissues but both the Pix2Pix GAN and the WGAN performed poorly for the boundaries and sometimes visual artifacts in the corner of the synthetic images. E-GAN shows a good performance in terms of generating boundaries and edges.

As generally done by other works in this area, we assumed three different metrics to quantitatively evaluate the proposed method: (I) Peak signal-to-noise ratio (PSNR), (II) mean absolute error (MAE), (III) the structural similarity (SSIM), and (IV) normalized cross-correlation (Norm. $X_{Corr}$). (Table 1).

To evaluate the effect of our model choices, we computed matrices values with different possible options. Table 2, represents the result of implementing different loss functions with the same architecture.

According to the experimental result presented in Table 2, E-GAN enhances PSNR to 22.13% compared to the mean PSNR in the state of the art. The generated images with E-GAN have higher SSIM, correlation coefficient and a lower MAE as well. The highest increment is a 67% increase in MAE compared to the mean MAE in the state of the art methods.

In Table 3, JSD and L2 loss shows improvement in the all metrics. For example, compared with the MSE the averaged PSNR of the proposed objective function increases by approximately 3.71%. The experimental results demonstrate that the adversarial training mechanism of GAN (KL divergence) is not always able to converge.

We believe that the geometrical and edges information can help the model to face the saturation problem less. This information helps the generator to be able to catch up with the discriminator faster. Figure 6, illustrates the synthetic images with geometrical information and without geometrical information.

### 5.2. Structural Analysis

Visual analysis is one of the most common ways to evaluate generated images. To ensure that the structural information is not missed, the generated images have been evaluated after segmenting the generated image as well. We segmented each image into three classes (white matter, grey matter, and CSF) to examine if the model has generated miss-placed voxels. Figure 7, illustrates the ground truth segmented T1 weighted MRI and the synthetic image after the segmentation.

The proposed model almost predicts the structure of the grey matter correctly so the generated grey matter can be used in order to correct the partial volume effect, on PET images. We have replicated the analysis with 10 subjects and measured the average voxel values between the ground truth and the generated images for each tissue. Figure 8, represents the average of voxel values.

As expected the difference in the voxel values between the generated images and real images is quite low. It means that the generated images regarding the voxels values are highly correlated to the ground truth. In the next section, we assess the model in terms of tissue analysis.

### 5.3. Tissue Analysis

Gray Level Co-occurrence Matrix (GLCM) is a common texture descriptors. GLCM calculates the Haralick features based on grey level intensities of the image. These features are useful in texture recognition, image segmentation, image classification, and object recognition [36]. In order to evaluate E-GAN in terms of simulating the texture, we have performed the GLCM on the generated images and the ground truth. To do so, each image has been discretized into eight grey levels and the Co-occurrence matrix has been computed for the state of the art methods and the proposed model. Figure 9, shows the Co-occurrence in 3D for a 6o degree angle and distance equal to three as an example. The Co-occurrence matrix of the proposed model is the most similar one to the ground truth Co-occurrence matrix. Pix2Pix GAN also represented a good performance to generate the texture descriptor. In order to have a quantitative assessment of the texture Table 4, represents a comparison of the four most common Haralick features from GLCM for the proposed architecture and the state of the art methods.

The energy of the generated image with E-GAN is equal to the real image so the model can generate the localized changes of the image as same as the real image. As shown in Table 4, the proposed model has less contrast error in the generated images compared to the other methods. Although the homogeneity is approximately the same between the different models, E-GAN has the minimum difference with the ground truth. E-GAN can generate images with the similar textures information to the real image. This is critical when the image-to-image translation will be done with the aim of diagnosing or generating abnormal images to balance a dataset.

## 6. Conclusions

In this paper, we proposed a model to generate 3D T1 weighted MRI from the FDG-PET images. E-GAN is capable of generating 3D T1 weighted MRI by eliciting relevant 3D features with separable convolutions and a Sobel filter. Moreover, the idea of adding a Sobel filter for passing the geometrical information of PET images through the decoder helps the generator to avoid the saturation problem in the discriminator by generating more realistic samples in the first steps of the training. Experimental results on the ADNI dataset show that the proposed model can defeat the state of the art and generate images of a better quality. The evaluation of the proposed model has been done in three aspects: visual analysis, structural analysis, and tissue analysis. Although E-GAN demonstrated good performance compared to the state-of-the-art methods, the operations that have to be done in the generator in order to mix the features increase the training time of the network, which could be a drawback. Another limitation is related to min–max strategy. Adding the Sobel filter in practice improves the stability of the network at the first steps of training but still sometimes the network faces saturation problem. The proposed model has only tested on healthy subjects. As the next step of this study, it could be beneficial to generate abnormal images such as images related to different stages of Alzheimer’s disease.

## Figures and Tables

**Figure 1 sensors-22-04640-f001:**
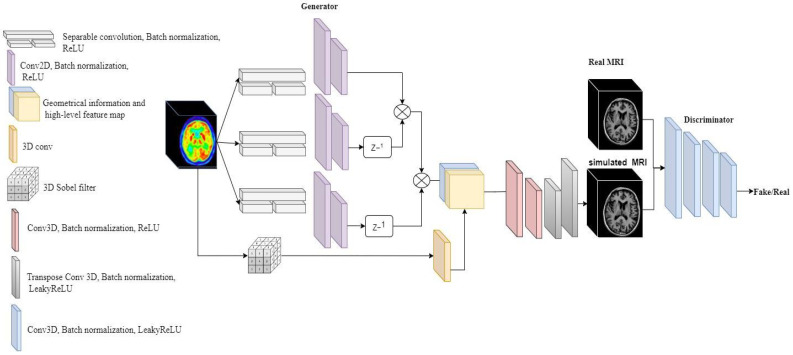
Network architecture of E-GAN for image-to-image translation.

**Figure 2 sensors-22-04640-f002:**
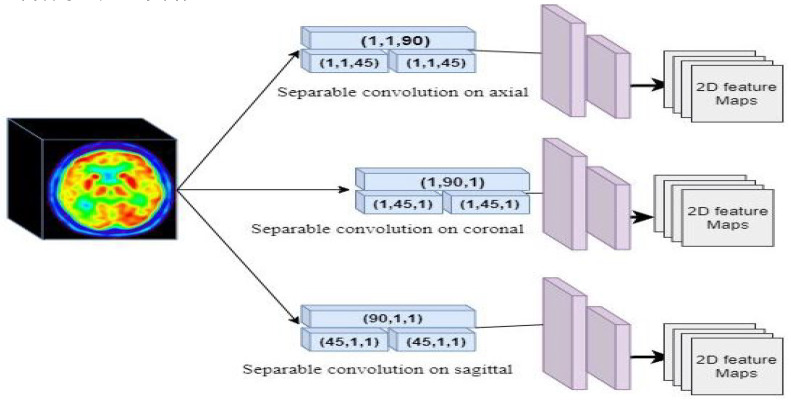
Configuration of the Elicit Network for projection the features in 2D.

**Figure 3 sensors-22-04640-f003:**
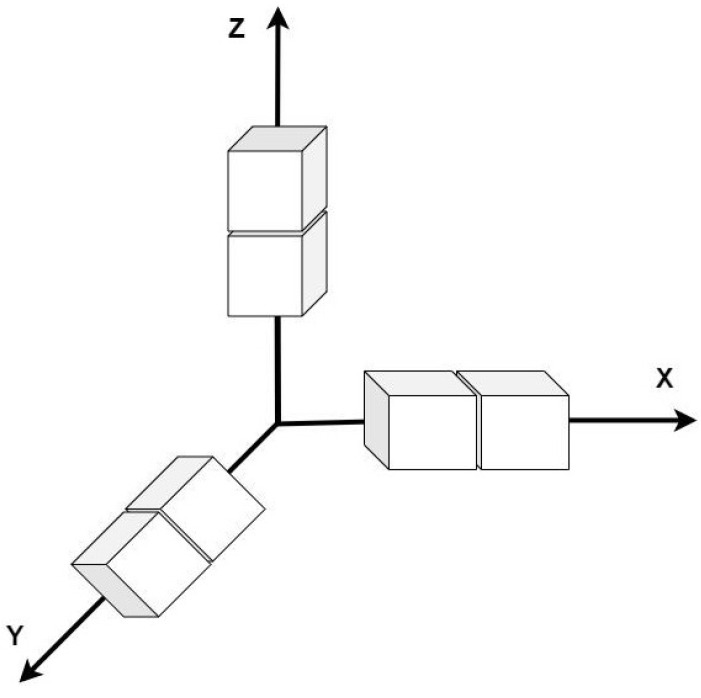
3D Sobel operation in x, y and z.

**Figure 4 sensors-22-04640-f004:**
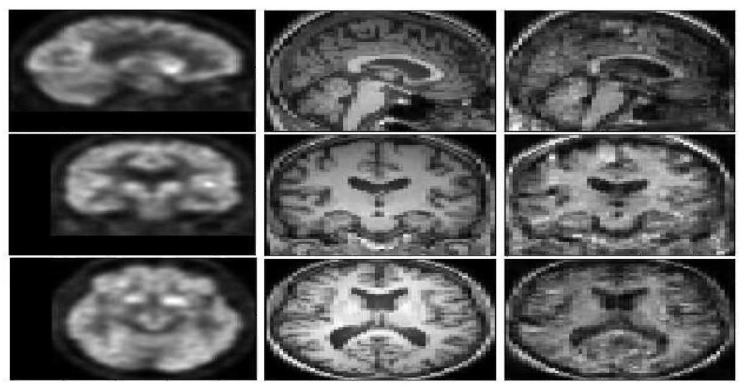
A pair of FDG-PET (left image), the ground truth T1 weighted MRI (middle image), and the generated image (right image) with E-GAN.

**Figure 5 sensors-22-04640-f005:**
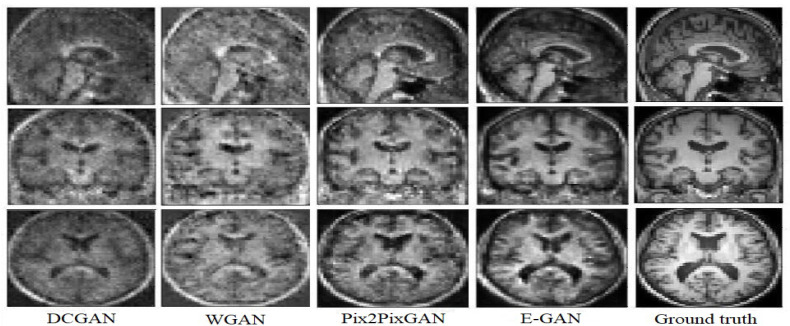
Experimental results for translating PET images to corresponding MRI with 3D DCGAN [35], 3D WGAN [25], 3D Pix2Pix GAN [17], and the proposed method.

**Figure 6 sensors-22-04640-f006:**
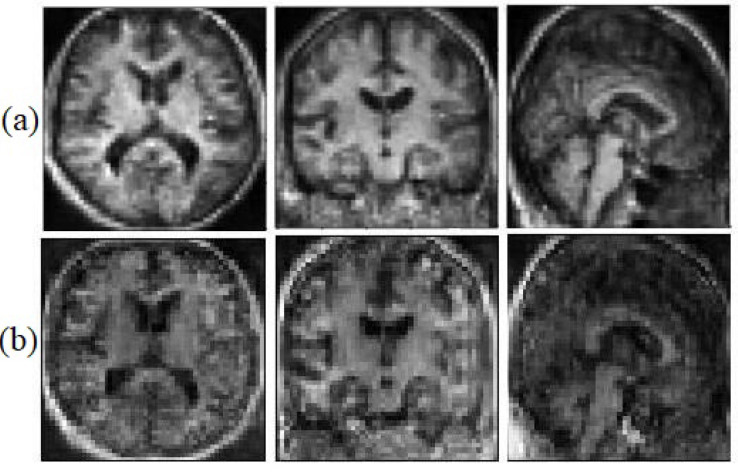
Synthetic MRI with the Sobel filter (**a**) and without the Sobel filter (**b**).

**Figure 7 sensors-22-04640-f007:**
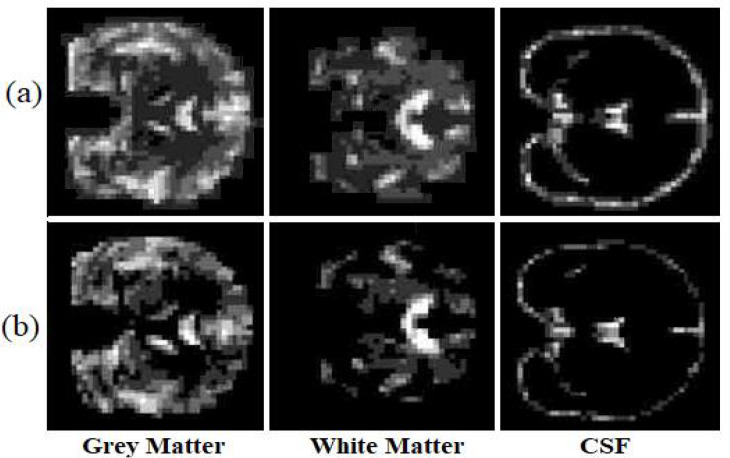
The ground truth (**a**) and the synthetic T1 weighted MRI (**b**).

**Figure 8 sensors-22-04640-f008:**
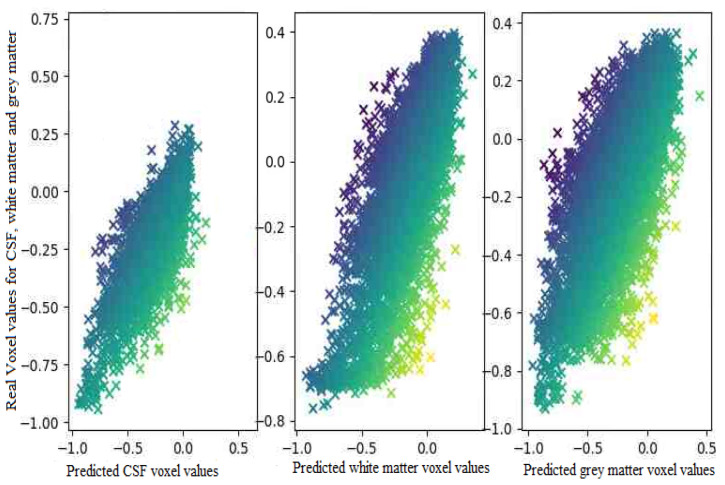
The Average voxel values in the translated image and the ground truth for 10 subjects.

**Figure 9 sensors-22-04640-f009:**
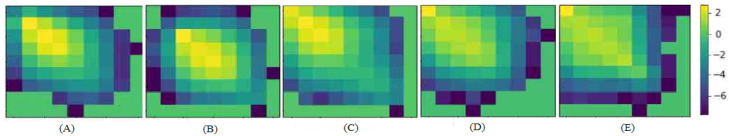
The Co-occurrence matrix DCGAN [35] (**A**), WGAN [25] (**B**), Pix2Pix GAN [17] (**C**), Proposed model (**D**) and the ground truth (**E**) in 3D.

**Table 1 sensors-22-04640-t001:** Strengths and weaknesses of the state of the art methods and the proposed method.

Method	2D/3D	Strengths	Weaknesses
Chartsias et al. [8]	2D	Generating Unpair images	Only task evaluation
Salman et al. [9]	2D	Deal with Source noise	
MustGAN [10]	2D	Aggregate the information from different modalities	Paired source-target images are Required
Ma et al. [12]	3D	Improve the generation of hard-to-synthesis voxels in the hippocampus regions	Discontinuities in the third axes
Ea-GANs [15]	3D	Improve the performance by adding edge similarity	Training a Complex architecture (U-net) with a few samples
E-GAN	3D	Reduces the complexity of the network	Training the network is time consuming

**Table 2 sensors-22-04640-t002:** The evaluation metrics of different state of the art methods on ADNI and the proposed method after 1000 epochs.

Model	PSNR	SSIM	MAE	$X_{\mathrm{Corr}}$
DCGAN [35]	21.4	0.53	724	0.604
WGAN [25]	23.01	0.58	445	0.645
Pix2Pix [17]	24.76	0.61	295	0.726
E-GAN	28.16	0.75	105	0.809

**Table 3 sensors-22-04640-t003:** The evaluation metrics of different loss functions on ADNI for the proposed method after 1000 epochs.

Loss	PSNR	SSIM	MAE	$X_{\mathrm{Corr}}$
MSE	25.49	0.602	110	0.733
KL	24.45	0.02	640	0.69
MSE+JSD	28.16	0.75	105	0.809

**Table 4 sensors-22-04640-t004:** Comparison of Haralick features for the generated images and the ground truth.

Model	Energy	Homogeneity	Dissimilarity	Contrast
Real image	0.734	0.692	0.716	1.971
E-GAN	0.734	0.664	0.826	1.835
Pix2Pix GAN [17]	0.371	0.611	1.010	1.755
WGAN [25]	0.276	0.671	0.555	1.244
DCGAN [35]	0.243	0.680	0.319	1.130

## Data Availability

Data used in preparation of this article were obtained on (10/11/2020) from the Alzheimer’s Disease Neuroimaging Initiative (ADNI) database (adni.loni.usc.edu). As such, the investigators within the ADNI contributed to the design and implementation of ADNI and/or provided data but did not participate in analysis or writing of this report. A complete listing of ADNI investigators can be found at http://adni.loni.usc.edu/wp-content/uploads/how_to_apply/ADNI_Acknowledgement_List.pdf.

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
