# Peer review of "FDG-PET to T1 Weighted MRI Translation with 3D Elicit Generative Adversarial Network (E-GAN)"

_sensors, 2022, doi:10.3390/s22124640_

Round 1

Reviewer 1 Report

In this paper, the authors have proposed the translation of FDG-PET to T1 weighted MRI with 3D Elicit-GAN. Overall paper is well written but requires some major improvements mentioned below. 

1) Please include the quantitative results in the abstract. 

2) The novelty of the work is limited. Please include a separate section for contributions and highlight the novelty of the work in at least 3-4 points.

3) In the related work, please include the comparison table that should highlight the strengths and weaknesses of the proposed as well as previous methods.

4) In figures and tables, please include the reference number of the previous method for easy understanding. 

5) Please mention the limitations of the work and also write future work in the conclusion section. 

Author Response

Point 1:  Please include the quantitative results in the abstract. 

Response 1: a sentence has been added to the Abstract (highlighted )

Point 2:  The novelty of the work is limited. Please include a separate section for contributions and highlight the novelty of the work in at least 3-4 points.

Response 2: The main contributions of this paper are as follows: I) We Split learning 3D features with separable convolution in a generative model instead of full convolution, II)  We propose a fusion strategy similar to the self-attention mechanism in order to put in order the extracted features, III) EGAN include a Sobel filter for transmission of the geometrical information instead of using 3D convolution. VI) We improved the stability of learning with a weighted version of a hybrid loss function.

Point 3: In the related work, please include the comparison table that should highlight the strengths and weaknesses of the proposed as well as previous methods.

Response 3: A table has been added at the end of the state of the art section

Point 4:  In figures and tables, please include the reference number of the previous method for easy understanding. 

Response 4: The reference numbers of the previous method have been included as follows.

  • DCGAN [38]
  • WGAN [26]
  • Pix2Pix [28]

Point 5:Please mention the limitations of the work and also write future work in the conclusion section.

Response 5: a paragraph has been added to the last part of the conclusion section.

Point 6: Does the introduction provide sufficient background and include all relevant references? can be improved 

Response 6: Please see the introduction section in the attachment 

Point 7: Are the methods adequately described? can be improved

Response 7: Please see the introduction section in the attachment 

Point 8: Are the result clearly presented? must be improved

Response 8: Please see the introduction section in the attachment 

Point 9: Are the conclusions supported by the results? Can be improved

Response 9: Please see the introduction section in the attachment 

Reviewer 2 Report

This manuscript reports the results of attempts to take PET images and make them into MR images using poorly-described computer tricks.  I found this difficult to read due to the rambling style and the many, many unsubstantiated claims.  The authors suggest something is "better" but don't tell with respect to what, or how.  Although this submission might be of interest to a handful of "deep learning" experts, it is not written in a manner that would appeal to a more general imaging audience.  

That said, my main reservation about this paper is this: Why would someone wish to take a PET image and make an MR image out of it in the first place?  How this would be of interest outside the math community, I can't even begin to guess.  In order to be of interest to a radiologist, they at least should show the input PET images, but not only do they not show them side by side, they don't even show the PET images at all so that it is impossible for the reader to judge the results.  Furthermore, I find no discussion of the fact that these two image types have nothing to do with each other, aside from being taken from the brains of the same subjects.  These two image types arise from completely different physical mechanisms; the intensity of a pixel in an MR image is governed by the proton density and the nuclear relaxation times, while that for the PET image simply reflects the amount of tracer in the tissue volume.  These different pixel intensities have nothing in common except the underlying anatomy.  In addition, no mention is made of the fact that the resolution scales are quite different between the two imaging methods.  MRI can achieve sub-mm resolution, while PET is limited by the positron diffusion distance to around 5 mm.  This paper would require substantial, in depth revision in order to constitute a valuable contribution to the imaging literature.

Author Response

Point 1:  

This manuscript reports the results of attempts to take PET images and make them into MR images using poorly-described computer tricks.  I found this difficult to read due to the rambling style and the many, many unsubstantiated claims.  The authors suggest something is "better" but don't tell with respect to what, or how.  Although this submission might be of interest to a handful of "deep learning" experts, it is not written in a manner that would appeal to a more general imaging audience.  

That said, my main reservation about this paper is this: Why would someone wish to take a PET image and make an MR image out of it in the first place?  How this would be of interest outside the math community, I can't even begin to guess.  In order to be of interest to a radiologist, they at least should show the input PET images, but not only do they not show them side by side, they don't even show the PET images at all so that it is impossible for the reader to judge the results.  Furthermore, I find no discussion of the fact that these two image types have nothing to do with each other, aside from being taken from the brains of the same subjects.  These two image types arise from completely different physical mechanisms; the intensity of a pixel in an MR image is governed by the proton density and the nuclear relaxation times, while that for the PET image simply reflects the amount of tracer in the tissue volume.  These different pixel intensities have nothing in common except the underlying anatomy.  In addition, no mention is made of the fact that the resolution scales are quite different between the two imaging methods.  MRI can achieve sub-mm resolution, while PET is limited by the positron diffusion distance to around 5 mm.  This paper would require substantial, in-depth revision in order to constitute a valuable contribution to the imaging literature

Response 1: 

In this study, we present a cross-modality generation method to generate T1 weighted MRI from Fluorine-18-Flourodeoxygloccose Positron Emission Tomography (FDG-PET). This dual information (PET+MRI) is not always available, especially in the same temporal window (If the two examinations are not done at the same time the two parameters change over time with a loss of information). Having the dual modalities (PET+MRI) for the inclusion of healthy subjects in scientific studies could be beneficial. Besides these aspects, the map of grey-matter density will allow correcting the partial volume effect on PET images and thus improve the quality of PET images. Despite this, the objective is not to entirely replace MRI with PET since MRI is multimodal imaging not limited to the measure of grey matter density. 

Point 2:  Does the introduction provide sufficient background and include all relevant references? Must be improved

Response 2: Please see the attachment 

Point 3: Are the methods adequately described? Must be improved

Response 3:  Please see the attachment 

Point 4: Are the results clearly presented? Must be improved

Response 4: Please see the attachment  

Point 5: Are the conclusions supported by the results? Must be improved

Response 5: Please see the attachment 

Round 2

Reviewer 1 Report

Most of my comments are addressed. I recommend the acceptance of this manuscript in its present form. 

Author Response

Does the introduction provide sufficient background and include all relevant references? Can be improved 

The motivations have been explained better in the introduction.

Reviewer 2 Report

Since the authors refused to consider the fact that MRI and PET are completely different methods, with differing contrast mechanisms and vastly differing response to brain anatomy, I must reject this paper.  For example, PET only shows the brain parenchyma, while MRI shows the skin, skull, white matter, grey matter, blood vessels, etc. and the resolution differs by at least an order of magnitude, I simply don't believe this work.  How can one take a 5 mm resolution image and make a 0.5 mm resolution image.  The data in the PET images simply  doesn't exist, no matter how many layers of computer nonsense one puts the PET image through.

Author Response

Point 1: Since the authors refused to consider the fact that MRI and PET are completely different methods, with differing contrast mechanisms and vastly differing responses to brain anatomy, I must reject this paper. For example, PET only shows the brain parenchyma, while MRI shows the skin, skull, white matter, grey matter, blood vessels, etc. and the resolution differs by at least an order of magnitude, I simply don't believe this work. How can one take a 5 mm resolution image and make a 0.5 mm resolution image. The data in the PET images simply don't exist, no matter how many layers of computer nonsense one puts the PET image through. 

Response 1: The fact that MRI and PET images are different hasn't been refused. The aim of this study is not to replace the MRI image with PET. To address the comments of the reviewer “The data in the PET images simply don't exist” we refer to a scientific study [Lin, Wanyun, et al. "Bidirectional mapping of brain MRI and PET with 3D reversible GAN for the diagnosis of Alzheimer’s disease." Frontiers in Neuroscience 15 (2021): 357.]
In the referenced study, a 3D reversible GAN has been proposed in order to generate MRI from PET and vice versa. In the conclusion of the paper, they mentioned “we can find that the structural and functional information of brain tissue can be mapped well, but it is difficult to map the structure information such as the skull of the MR image from the PET image” therefore the information does exist but mapping the structure of MRI is difficult especially the skull (“the accuracy of diagnosing is mainly based on the brain tissue area in the neuroimaging and is not sensitive to the skull and other structures”. 
The reference study has been done for four stages of Alzheimer's disease data from subjects in three categories: cognitively unimpaired (labelled as CN), mild cognitive impairment (MCI) and AD. As also mentioned in the reference article “In this study, the relationship between MRI and PET is a more complex non-linear complementary relationship” it is more difficult to map the information from the PET image to the MRI image because of the variability of the disease.
 In our study, we only focused on healthy subjects which is simpler to predict the brain tissues (the main aim of our study is mapping the grey matter not the skull as mapping the skull is hard from the PET image) 

Please see the attachment too.
